# Rationale-Enhanced Language Models are Better Continual Relation Learners

**Weimin Xiong    Yifan Song    Peiyi Wang    Sujian Li**[*]

National Key Laboratory for Multimedia Information Processing,
School of Computer Science, Peking University
{wmxiong,yfsong,lisujian}@pku.edu.cn
wangpeiyi9979@gmail.com

## Abstract

Continual relation extraction (CRE) aims to solve the problem of catastrophic forgetting when learning a sequence of newly emerging relations. Recent CRE studies have found that catastrophic forgetting arises from the model's lack of robustness against future analogous relations. To address the issue, we introduce rationale, i.e., the explanations of relation classification results generated by large language models (LLM), into CRE task. Specifically, we design the multi-task rationale tuning strategy to help the model learn current relations robustly. We also conduct contrastive rationale replay to further distinguish analogous relations. Experimental results on two standard benchmarks demonstrate that our method outperforms the state-of-the-art CRE models. Our code is available at https://github.com/WeiminXiong/RationaleCL

---

**Question:** It is the fourth installment in the "Predator" film series, set to take place between "*Predator 2*" (1990) and "*Predators*" (2010). Classify the relation in above sentence.
**Answer:** follow by

---

**Prompt:** The relation between "*Predator 2*" and "*Predators*" is "followed by". Please explain why.
**Rationale:** The relation between "*Predator 2*" (1990) and "*Predators*"(2010) is "followed by" because "*Predator 2*" is the second installment of the "Predator" film series, and it is followed by the fourth installment, which is "*Predators*".

---

**Similar relations:** follows; followed by; characters
**Prompt:** The relation between "*Predator 2*" and "*Predators*" is "followed by", instead of "follows" or "characters". Please explain why.
**Contrastive Rationale:** The sentence states that "*Predator 2*" took place before "*Predators*", so the proper relationship to denote this would be "followed by". The phrase "follows" suggests that "*Predator 2*" took place after "*Predators*". The phrase "characters" has no relation to the two films, as it refers to the people in a movie.

Figure 1: Examples of the input question text and the rationale and contrastive rationale for the relation answer.

## 1 Introduction

Relation extraction (RE) aims to identify the relations between two entities in a text. While traditional RE models cannot handle the real-life situation where new relations are constantly emerging, continual relation extraction (CRE) attempts to learn new relations while retaining the performance on learned relations (Han et al., 2020; Cui et al., 2021; Zhao et al., 2022). Similar to other continual learning tasks, the main challenge in CRE is the phenomenon of *catastrophic forgetting* (CF), i.e., the performance on identifying old relations degrades significantly while learning new relations.

Most previous CRE researches have attributed catastrophic forgetting to the destruction of representations learned on previous tasks when learning new tasks (Han et al., 2020; Cui et al., 2021; Wang et al., 2022a). They focused on recovering the representations on previous tasks, using methods like

restricted gradient updating and knowledge distillation (Lopez-Paz and Ranzato, 2017; Cao et al., 2020). Recently, another line of work (Wang et al., 2022b) found that in CRE scenario, models trained for the current task do not have good identification ability for new-coming relations which are analogous to a current relation. Thus, making a model learn current relations robustly to avoid subsequent confusion becomes the new research focus (Wang et al., 2022b; Zhao et al., 2023).

To address this problem, in this paper, we assume that incorporating rationales can enhance the performance of CRE models in learning analogous relations. This is inspired by the intuition that, training models with explicit rationale supervision can provide greater robustness (Chen et al., 2022). Moreover, since relation extraction requires reasoning over two entities, providing explanations for why the two entities have a specific relation can enhance the reasoning capacity of smaller models,

---
[*]Corresponding Author

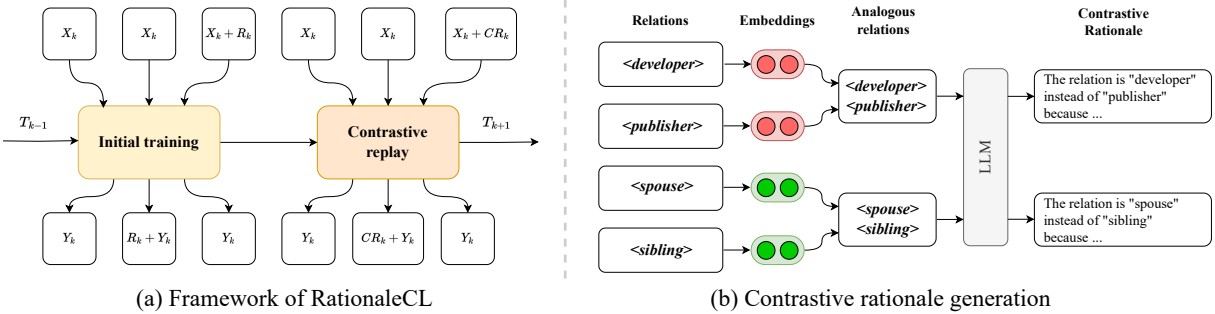

(a) Framework of RationaleCL      (b) Contrastive rationale generation

Figure 2: An overall demonstration of our proposed RationaleCL.

thereby eliminating reliance on spurious shortcuts (Li et al., 2022; Magister et al., 2023). As there are no such corpus which labels the entity relations with corresponding rationales, we propose to make use of LLM to generate the explanations, i.e., the rationales, for the relation classification answers with prompting questions, as shown in Figure 1.

To fully exploit rationales, we propose a novel approach called **RationaleCL**, which incorporates two strategies: multi-task rationale tuning and contrastive rationale replay, into a rehearsal-based framework, as shown in Figure 2. Specifically, we employ the encoder-decoder model T5 (Raffel et al., 2020) as our backbone, utilizing multi-task rationale tuning with three tasks: question to answer as the main task, question to rationale-answer and question-rationale to answer as auxiliary tasks. With the rationale tuning strategy, we distill the rationale knowledge from LLM to make T5 develop the reasoning ability to interpret its classification results, leading to enhanced robustness for the CRE task. When conducting memory rehearsal, we prompt LLM to differentiate between analogous relations and regenerate the corresponding explanations, i.e., the contrastive rationales (Figure 1), to update the memory. which not only helps mitigate catastrophic forgetting but prevents confusion.

Our contributions are summarized as follows: (1) For the first time, we introduce rationale generated by LLM into CRE task to mitigate catastrophic forgetting. (2) We propose a novel rationale-enhanced CRE method RationaleCL, which incorporates multi-task rationale tuning and contrastive rationale replay strategies. (3) Experimental results on FewRel and TACRED verify the effectiveness of our method.

## 2 Task Formalization

In this work, we focus on continual learning for a sequence of $n$ relation classification tasks $\{T_1, T_2, ..., T_n\}$. For task $T_k$, given a training set $D_k = \{\langle x_i, y_i \rangle | x_i \in X_k, y_i \in Y_k\}$, the model learns to identify the corresponding relation type $y_i$ for the input text $x_i$, where $Y_k$ denotes a set of new relation types and $X_k$ a set of text containing two entities. The goal of CRE is to continually train the model on new tasks to learn new relations, while avoiding forgetting of previously learned ones. Therefore, after learning $n$ tasks, the model will be evaluated to identify each text $x$ from test set into a relation type $y \in \cup_{i=1}^{k} Y_i$.

## 3 Methodology

Figure 2 shows the model architecture of RationalCL, which follows a two-stage training paradigm. In *Stage 1*, we first make use of LLM to generate rationales, with which the T5 model is trained under the multi-task rationale tuning framework to learn to identify new relations. Then, to alleviate catastrophic forgetting in CRE, we adopt an episodic memory module to store a few representative instances for each relation. The memory data will be continually learned with emerging new tasks. When selecting these representative instances for each relation, we follow Han et al. (2020) and use the K-means algorithm to conduct clustering based on the features of all samples extracted by T5-encoder, and select the samples in the centre of each cluster into memory. In *Stage 2*, we regenerate the contrastive rationale and conduct contrastive rationale replay under the multi-task framework to alleviate catastrophic forgetting.

### 3.1 Rationale Generation

Previous researches have illustrated the capability of LLM to decompose problems into a series of

| TACRED | | | | | | | | | | |
|---|---|---|---|---|---|---|---|---|---|---|
| **Models** | **T1** | **T2** | **T3** | **T4** | **T5** | **T6** | **T7** | **T8** | **T9** | **T10** |
| RPCRE (Cui et al., 2021) | 97.6 | 90.6 | 86.1 | 82.4 | 79.8 | 77.2 | 75.1 | 73.7 | 72.4 | 72.4 |
| EMAR (Han et al., 2020) | 97.8 | 92.4 | 89.6 | 84.6 | 83.2 | 81.3 | 78.7 | 77.1 | 77.3 | 76.8 |
| CRECL (Hu et al., 2022) | 97.3 | 93.6 | 90.5 | 86.1 | 84.6 | 82.1 | 79.4 | 77.6 | 77.9 | 77.4 |
| CRL (Zhao et al., 2022) | 97.7 | 93.2 | 89.8 | 84.7 | 84.1 | 81.3 | 80.2 | 79.1 | 79.0 | 78.0 |
| ACA (Wang et al., 2022b) | 98.0 | 92.1 | 90.6 | 85.5 | 84.4 | 82.2 | 80.0 | 78.6 | 78.8 | 78.1 |
| CEAR (Zhao et al., 2023) | 97.7 | 94.3 | **92.3** | **88.4** | **86.6** | 84.5 | 82.2 | 81.1 | 80.1 | 79.1 |
| RationaleCL | **98.6** | **94.4** | 91.5 | 88.1 | 86.5 | **84.9** | **84.5** | **82.5** | **81.6** | **80.8** |

| FewRel | | | | | | | | | | |
|---|---|---|---|---|---|---|---|---|---|---|
| **Models** | **T1** | **T2** | **T3** | **T4** | **T5** | **T6** | **T7** | **T8** | **T9** | **T10** |
| RPCRE (Cui et al., 2021) | 97.9 | 92.7 | 91.6 | 89.2 | 88.4 | 86.8 | 85.1 | 84.1 | 82.2 | 81.5 |
| EMAR (Han et al., 2020) | 98.2 | 94.1 | 92.0 | 90.8 | 89.7 | 88.1 | 87.2 | 86.1 | 84.8 | 83.6 |
| CRECL (Hu et al., 2022) | 98.0 | 94.7 | 92.4 | 90.7 | 89.4 | 87.1 | 85.9 | 85.0 | 84.0 | 82.1 |
| CRL (Zhao et al., 2022) | 98.1 | 94.6 | 92.5 | 90.5 | 89.4 | 87.9 | 86.9 | 85.6 | 84.5 | 83.1 |
| ACA (Wang et al., 2022b) | 98.3 | 95.0 | 92.6 | 91.3 | 90.4 | 89.2 | 87.6 | 87.0 | 86.3 | 84.7 |
| CEAR (Zhao et al., 2023) | 98.1 | **95.8** | **93.6** | 91.9 | 91.1 | 89.4 | 88.1 | 86.9 | 85.6 | 84.2 |
| RationaleCL | **98.6** | 95.7 | 93.4 | **92.3** | **91.3** | **89.7** | **88.2** | **87.3** | **86.3** | **85.1** |

Table 1: Accuracy (%) on all seen relations after learning each task. We show the best results in **boldface**.

intermediate reasoning steps[1] (Wei et al., 2023; Kojima et al., 2023). Therefore, we leverage this capability of LLM to produce explanation, i.e., rationale, for the relation classification datasets. Specifically, as shown in Figure 1, for each instance $\langle x_i, y_i \rangle$ in the training set, we construct the corresponding prompt and let the LLM (*gpt-3.5-turbo* in this paper) generate the rationale $r_i$ explaining why $x_i$ is identified as the relation type $y_i$[2].

### 3.2 Multi-task Rationale Tuning

Given an instance $\langle x_i, y_i \rangle \in D_k$ and its rationale $r_i$, we define three tasks: question to answer ($\text{Task}_c$), question to rationale-answer ($\text{Task}_r$) and question-rationale to answer ($\text{Task}_d$) as follows:

$$\text{Task}_c : x_i \rightarrow y_i$$
$$\text{Task}_r : x_i \rightarrow r_i + y_i$$
$$\text{Task}_d : x_i + r_i \rightarrow y_i$$

Figure 2(a) illustrates the multi-task rationale tuning strategy. $\text{Task}_c$ is the main task, which let the model directly generate the relation $y_i$ for the input question text $x_i$. To enhance the robustness of the model for future analogous relations, we design two auxiliary tasks $\text{Task}_r$ and $\text{Task}_d$ to help develop its reasoning ability to interpret its

classification results. $\text{Task}_r$, which is similar to Chain-of-Thought reasoning, instructs the model to predict the relation classification result $y_i$ and generate the corresponding rationale $r_i$ why $y_i$ is predicted from the question $x_i$. This task helps the model grasp the real rationale and thereby avoid dependency on the spurious shortcut. To enable the model make full use of the rationale for relation prediction, $\text{Task}_d$ is designed to take $x_i$ and $r_i$ as input and produce a relation result $y_i$.

The training objective of our model is a combination of the losses for the three tasks above:

$$\mathcal{L} = \alpha \mathcal{L}_c + (1 - \alpha)(\beta \mathcal{L}_r + (1 - \beta)\mathcal{L}_d),$$

where $\alpha$ and $\beta$ are factors used to adjust the loss weights.

### 3.3 Contrastive Rationale Replay

In the replay stage, the model has access to memory from all previously learned relations. To further reduce confusion over analogous relations, we prompt LLM to regenerate the contrastive rationale that distinguishes them, and update memory with the new rationale. The process of contrastive rationale generation is depicted in Figure 2(b). As shown, we first find the analogous relations for each relation. Following Wang et al. (2022b), we employ the cosine distance of the average embedding of the instances as a metric to measure the similarity of two relations. We consider two relations

---

[1]However, LLM faces difficulties in tackling the RE problem involving numerous relations on its own without finetuning. Please refer to Appendix B for more detail.

[2]We save the generated rationale for re-using.

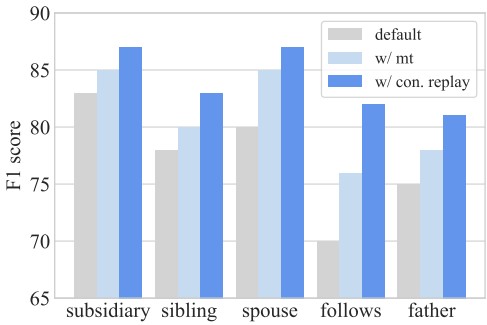

Figure 3: Performance on analogous relations. "mt" denotes multi-task rationale tuning. "con. replay" denotes contrastive rationale replay.

are similar if their similarity exceeds a threshold $\tau$. For each instance $\langle x_i, y_i, r_i \rangle$ stored in memory, we identify the similar relations of $y_i$ and create a new prompt to regenerate its contrastive rationale $cr_i$, as shown in Figure 1. The contrastive rationale $cr_i$ is intended to highlight the distinctions between the similar relations and is used to replace $r_i$ in memory. Subsequently, the model is trained on the updated memory.

## 4 Experiments

### 4.1 Experimental Settings

**Datasets** Following previous work (Han et al., 2020; Cui et al., 2021; Hu et al., 2022; Wang et al., 2022a), our experiments are conducted upon two widely-used datasets, **FewRel** (Han et al., 2018) and **TACRED** (Zhang et al., 2017). Please refer to Appendix A.1 for more details.

**Implementation Details** To ensure fair comparisons, we adhere to the identical experimental setting employed by (Wang et al., 2022a). We randomly divide all relations into 10 subsets corresponding to 10 tasks, and *accuracy* on all observed relations is chosen as the evaluation metric. The random seeds are identical to guarantee that the task sequences remain the same. We maintain a fixed number of 10 stored instances in memory for each relation and report the average result of five different task sequences. More details of our experimental settings and comparison baselines are included in Appendix A.2 and A.3.

### 4.2 Main Results

The performances of our proposed RationaleCL and baselines on two datasets are shown in Table 1. Our method consistently outperforms all baselines with significance test $p < 0.05$. For the accuracy of

T10, RationaleCL improves the accuracy of SOTA methods CEAR/ACA by 1.7/2.7 and 0.9/0.4 on TACRED and FewRel, respectively. These results demonstrate the effectiveness and universality of our proposed method.

### 4.3 Analysis

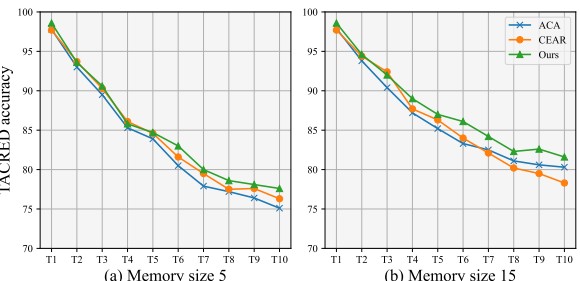

Figure 4: Comparison of model's performance on different memory sizes.

**Improvements on Analogous Relations** We further analyse the ability of the model to distinguish between analogous relations. We show the performance of relations which has larger similarity to other relations, as mentioned by (Wang et al., 2022b). As shown in Figure 3, the model's F1 scores on those relations are improved greatly after using multi-task rationale tuning and further enhanced with contrastive rationale replay. We show some cases in Appendix D.

**Influence of Memory Size** Memory size is defined as the number of stored typical samples for each relation, which is a key factor for the model performance of rehearsal-based CRE methods. Therefore, we study the influence of memory size on our method. As shown in Figure 4, our method outperforms ACA and CEAR with memory sizes 5 and 15, demonstrating that our model is robust to the change of memory size.

**Ablation Study** We further conduct an ablation study of our proposed two strategies. The experimental results in Table 2 show a performance degradation with the ablation of both strategies, demonstrating the effectiveness of our proposed rationale-enhanced framework. For more ablation results, please refer to Appendix C.

## 5 Conclusion

In this paper, we devise a framework by introducing rationales generated by LLM into CRE tasks to improve the model's robustness against future

| Models | TACRED | FewRel |
|--------|--------|--------|
| RationaleCL | **80.8** | **85.1** |
| w/o con. replay | 80.2 | 84.8 |
| w/o $Task_d$ | 80.0 | 84.6 |
| w/o both | 79.8 | 83.9 |

Table 2: Ablation study results. "con. replay" denotes contrastive rationale replay.

analogous relations. Our method incorporates two strategies to help the model effectively learn current relations and better differentiate between analogous relations. Experimental results on two benchmarks show that our method consistently outperforms previous state-of-the-art CRE models. Further analysis confirms the effectiveness of our proposed rationale-enhanced framework.

## Limitations

Our paper has several limitations: 1) Our method uses manually designed prompts to generate rationales. However, the choices of prompts may have a great impact on the quality of rationales, which has not been investigated. 2) Our method suffers from efficiency problems. On the one hand, the multi-task rationale tuning strategy increases GPU memory consumption and introduces extra computational overhead. On the other hand, the generation of contrastive rationale needs to be carried out repetitively, increasing the consumption of calling LLM API. 3) While our method is developed for the CRE task, it can also be applied to other continual learning tasks, which will be a focus of our future work.

## Ethics Statement

Our work complies with the ACL Ethics Policy. Since relation classification is a standard task in NLP and all datasets we used are publicly available, we have not identified any significant ethical considerations associated with our work.

## Acknowledgement

We thank the anonymous reviewers for their helpful comments on this paper. This work was partially supported by National Key R&D Program of China (No. 2022YFC3600402) and National Social Science Foundation Project of China (21&ZD287).

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

## A Experimental Details

### A.1 Datasets

Following previous work (Han et al., 2020; Cui et al., 2021; Hu et al., 2022; Wang et al., 2022a), our experiments are conducted upon the following two standard benchmarks with the train-test-validation split ratio set to 3:1:1.

**FewRel** (Han et al., 2018) It is a RE benckmark dataset originally proposed for few-shot learning. The dataset contains 100 relations, each with 700 instances. Following the previous work (Wang et al., 2019; Han et al., 2020), we use the original training and validation set of FewRel, which contains 80 relations.

**TACRED** (Zhang et al., 2017) It is a large-scale RE dataset containing 42 relations (including *no_relation*) and 106,264 samples, which is constructed on news networks and online documents. Following (Cui et al., 2021), *no_relation* was removed in our experiments, and the number of training samples for each relation is limited to 320 and the number of test samples of each relation to 40.

### A.2 Experimental Details

In our experiment, we use T5-base-lm-adapt as our backbone model and Adam as our optimizer. We set the learning rate 1e-4 for the model. The batch size of training is 32 and 16 for FewRel and TACRED respectively. The memory size of each task is 10. The training epoch for stage 1 and stage 2 are set to 10. Our experiments are conducted on a single NVIDIA A800 GPU.

We find the best hyperparameter values through grid search with a step of 0.1 for $\alpha$, $\beta$ and $\tau$. The search spaces for various hyperparameters are $\alpha \in [0.4, 0.9]$, $\beta \in [0.4, 0.6]$ and $\tau \in [0.95, 0.99]$. The used hyperparameter values are listed below:

- For FewRel, $\alpha = 0.6, \beta = 0.5, \tau = 0.97$.

- For TACRED, $\alpha = 0.9, \beta = 0.5, \tau = 0.97$.

### A.3 Baselines

We compare our proposed framework with the following baselines in our experiments:

- **EMAR** (Han et al., 2020) constructs a memory activation and reconsolidation mechanism to alleviate the catastrophic forgetting.

| Datasets | Test set sizes | Accuracy |
|----------|----------------|----------|
| FewRel   | 11200          | 38.71    |
| TACRED   | 1240           | 54.84    |

Table 3: Performance of LLM in FewRel and TACRED.

- **RPCRE** (Cui et al., 2021) proposes a relation prototypes and a memory network to refine sample embeddings, which effectively retains the learned representations in CRE.

- **CRL** (Zhao et al., 2022) proposes to utilize contrastive learning and knowledge distillation to alleviate catastrophic forgetting.

- **CRECL** (Hu et al., 2022) introduces prototypical contrastive learning to ensure that data distributions of all CRE tasks are more distinguishable to alleviate catastrophic forgetting.

- **ACA** (Wang et al., 2022b) designs two adversarial class augmentation mechanisms to learn robust representations to alleviate catastrophic forgetting.

- **CEAR** (Zhao et al., 2023) proposes memory-intensive relation prototypes and memory augmentation to reduce overfitting to typical samples in rehearsal stage.

## B Performance of LLM in RE tasks

As aforementioned in Section 3.1, although the LLM can generate a high-quality rationale for a given instance with its corresponding relation type, it faces difficulties in tackling the relation classification problem involving numerous relations on its own without finetuning. In this part, we conduct experiments to show the performance of LLM (*gpt-3.5-turbo* in this paper) in two RE tasks: FewRel and TACRED.

We prompt LLM to generate the relation type $y$ for an input text $x$ in zero-shot setting and restrict it to generate answers only within a given scope. We accomplish it by appending the set of relation types $Y$ to the input text $x$ and instructing LLM to select one best answer from $Y$. When an answer outside $Y$ is encountered, we discard it and keep regenerating until an answer in the set $Y$ is generated. The number of relation types contained in the set $Y$ is 80 and 40 corresponding to FewRel and TACRED, respectively.

| Models | TACRED | FewRel |
|---|---|---|
| RationaleCL | **80.8** | **85.1** |
| w/o Task$_r$-$cr$ | 80.6 | 84.9 |
| w/o Task$_d$-$cr$ | 80.3 | 84.9 |
| w/o Task$_r$-$cr$ & Task$_d$-$cr$ | 80.2 | 84.8 |

Table 4: More ablation study results.

As shown in Table 3, LLM performs poorly when prompted to directly generate the relation answer without finetuning. Therefore, we could not directly use LLM to perform RE tasks, but we could exploit its strong reasoning ability to provide rationale for a correct relation answer.

The poor performance of ChatGPT on relation extraction tasks has also been verified in several previous works (Li et al., 2023; Han et al., 2023). According to their studies, ChatGPT is limited by the output format requirements in accomplishing fine-grained relation extraction tasks, and it is difficult to directly generate the target relation label within the defined range. However, ChatGPT's semantic understanding is sufficient, and when we provide chatgpt with correct relation labels, ChatGPT can understand the meaning of the relation according to the context and give reasonable rationales. According to the human-check results (Li et al., 2023), domain experts highly approve of the reasons given by ChatGPT.

## C More Ablation Results

The purpose of contrastive replay is to produce contrastive rationale $cr_i$ and replace rationale $r_i$ in memory, which is involved in Task$_r$ and Task$_d$. As $r_i$ is replaced with $cr_i$ in both the tasks, there would be two "new" tasks for $cr_i$. We denote them as Task$_r$-$cr$ and Task$_d$-$cr$. In Task$_r$-$cr$, we only replace the rationale $r_i$ with $cr_i$ in Task$_r$ when conducting memory rehearsal while the opposite in Task$_d$-$cr$. We provide further ablation analysis in Table 4.

Conclusively, the ablation study underscores the importance of both Taskd-cr and Taskr-cr components in enhancing the performance of the RationaleCL model. Their presence contributes to the model's robustness to analogous relations, indicating their roles in tackling the catastrophic forgetting challenges of continual learning effectively.

## D Case Study

In this section, we present several cases where the original model without rationale tuning makes incorrect predictions, while RationaleCL correctly predicts the answer and the rationale, highlighting its robustness to analogous relations and avoidance of spurious shortcuts. These cases are shown in Table 5.

| Sentence with Instruction | Rationale Generated by Our Model | WA | RA |
|---|---|---|---|
| Given the subject entity "Lonely" and object entity "Belly Dancer (Bananza)", what is the relation type between them in sentence: It spawned the singles "Locked Up" and "Lonely", "Belly Dancer (Bananza)", "Pot Of Gold", and "Ghetto"? | The relation between "Lonely" and "Blody Dancer (Banza)" is "followed by" because "Lonely" is mentioned first in the sentence, followed by "Blody Dancer (Banza)" in the sentence. This indicates that "Blody Dancer (Banza)" is one of the singles that came after "Lonely" in the genre of music. | follows | followed by |
| Given the subject entity "ABC" and object entity "Sydney", what is the relation type between them in sentence: Hitchener moved to the Nine Network after one year at ABC Sydney, presenting "National Nine News" first in Sydney and then in Melbourne as the presenter of "News Centre Nine"? | ABC Sydney is a television station that specializes in broadcasting to the Sydney area. Hitchener moved to the Nine Network after one year at ABC Sydney, indicating that the station has its headquarters in Sydney. This indicates a relationship between ABC and Sydney of a "headquarters location". | licensed to broadcast to | headquarters location |
| Given the subject entity "Bolshevik revolution" and object entity "Petrograd", what is the relation type between them in sentence: They were attempting to flee from their home in Petrograd (St Petersburg) in Russia to escape the Bolshevik revolution, which Belevitch's father opposed? | The relation between "Bolshevik revolution" and "Petrograd" in the sentence is "location". This is because the sentence states that the people were attempting to flee from their home in Petrograd(St Petersburg) in Russia to escape the Bolshevik revolution. The Bolshevik revolution took place in Petrograd, which is located in Russia. | headquarters location | location |
| Given the subject entity "Kronos Quartet" and object entity "Nonesuch", what is the relation type between them in sentence: It was first recorded by guitarist Pat Metheny in 1987 and released together with Reich's "Different Trains", performed by the Kronos Quartet, on Nonesuch's 979 176 - 2? | The Kronos Quartet performed together with Reich's "Different Trains" on Nonesuch's 979 176-2, indicating that Nonesuch is the record label for the Kronos Quartet. This means that Nonesuch is the company responsible for distributing and promoting the Kronos Quartet's music on the record label. Therefore, the answer is: record label. | performer | record label |
| Given the subject entity "Monkey Island" and object entity "LucasArts", what is the relation type between them in sentence: "DeathSpank" features advanced dialog options, similar to LucasArts' "Monkey Island" series which play a large part in unfolding the game's story? | LucasArts is the publisher of the "Monkey Island" series of video games. It is responsible for the creation, promotion, and distribution of the series, making itself the publisher. Therefore, the answer is: publisher. | developer | publisher |

Table 5: Case Study. "Sentence with Instruction" denotes the input question sentence. "Rationale Generated by Our Model" denotes the rationale generated by our rationale-enhanced model. "WA" denotes the wrong answer generated by model without rationale tuning. "RA" denotes the right answer generated by our rationale-enhanced model. We highlight the reasoning process in rationale in blue.