# OpenReview forum: "Rationale-Enhanced Language Models are Better Continual Relation Learners"
_EMNLP/2023/Conference — EMNLP 2023 Main_

### Official Review · Reviewer_ergu · 2023-08-03

**Typos Grammar Style And Presentation Improvements:** 1) In Figure 2a, it would be helpful …
**Soundness:** 3

**Excitement:**

3: Ambivalent: It has merits (e.g., it reports state-of-the-art results, the idea is nice), but there are key weaknesses (e.g., it describes incremental work), and it can significantly benefit from another round of revision. However, I won't object to accepting it if my co-reviewers champion it.

**Paper Topic And Main Contributions:**

This paper introduces an approach for continual relation learning in language models. In particular, the method uses a LLM to generate rationales in order to learn relations. Their method involves two stages. The first stage involves training a language model on rationale-enhanced samples (generated by another LLM, which is GPT-3.5 in this paper) to facilitate the learning of relations. The second stage involves further training the language model on contrastive examples (also generated by GPT-3.5 in this paper) from memory between similar relations.

Empirical results show improvements over baselines on two relation extraction datasets: FewRel and TACRED. Additional analysis show improvements on specific cases of similar relations (e.g., “spouse”, “sibling”) and ablation studies show degradation with the removal of one or both stages of the method.


**Questions For The Authors:**

Question A: How is the loss for $Task_r$ (question to rationale-answer) calculated? It would be good to explicitly write the functions for $L_c$, $L_r$, and $L_d$ for clarity.

Question B: In Figure 3, how similar are the relations with respect to the metric they were calculated (i.e., cosine similarity)? Is it possible to plot a list of all relations with their similarity and f1 scores to observe trends in model performance as relation similar increases/decreases?

**Reasons To Accept:**

1) The paper demonstrates how rationale-tuning can be used for continual relation learning. In addition, given generation capabilities of LLMs, studying the effectiveness of using LLMs to augment data for such tasks is valuable.

2) The paper is clearly written and the methodology is explained well.

**Reasons To Reject:**

1) Improvements over other models and baselines are low. While ablation studies and analysis on specific examples of relations (e.g., “sibling”, “spouse”) as mentioned, a stronger argument for the effectiveness of the method could be made with additional analysis. For example, in what specific cases is the method most effective? Are there further trends that can be observed among these cases?

2) There is no analysis regarding the quality of generated rationales. Was there a post-processing step to filter invalid rationales? This information would be good to include in the methodology for paper soundness.

**Reproducibility:**

3: Could reproduce the results with some difficulty. The settings of parameters are underspecified or subjectively determined; the training/evaluation data are not widely available.

**Reviewer Confidence:**

4: Quite sure. I tried to check the important points carefully. It's unlikely, though conceivable, that I missed something that should affect my ratings.

---

> ### Author Rebuttal · Authors · 2023-08-26
>
> Thank you for your feedback and helpful suggestions on our work. We have carefully answered the questions in your comments.
> ## Q1: How is the loss for $Task_r$ calculated?
> The loss for $Task_r$ is calculated as $L_r = -log P(Y_k|concat(X_k, R_k))$, which represents negative log likelihood of the probability of generating the correct relation given the input text and rationale.
> ## Q2: How is the similarity between relations calculated?
> As mentioned in Line 168-171, we employ the cosine distance of the average embedding of the instances as a metric to measure the similarity of two relations[1].
> ## Q3: Improvements over other models and baselines are low.  Is there a stronger argument for the effectiveness of the method?
> To further verify the effectiveness of our method, we analyse the correlation between F1 score improvements (compared with default setting w/o both) per relation and its similarity with other relations. We split all relations into 10 groups according to its maximum similarity with other relations. The similarity ranges from 0.90 to 0.99 for ten groups, each group is spaced 0.1. We calculate the average of the F1 increases for all relations in each group as the increase for the whole group.
>
> **FewRel**
>
> |  Similarity      | 0.90 | 0.91 | 0.92 | 0.93 | 0.94 | 0.95 | 0.96 | 0.97 | 0.98 | 0.99 |
> | ----------- | ----------- | ----------- |----------- |----------- |----------- |----------- |----------- |----------- |----------- |----------- |
> | F1 improvements    | -0.45       | 0.37 |  0.41 | 0.23| 0.22 | 0.59 | 0.86 | 1.63 | 1.72 | 2.38 |
>
> Then we calculate the pearson correlation coefficient between similarity and F1 improvements. The result is 0.944, which indicates their strong correlation。
>
> **TACRED**
>
> |  Similarity      | 0.90 | 0.91 | 0.92 | 0.93 | 0.94 | 0.95 | 0.96 | 0.97 | 0.98 | 0.99 |
> | ----------- | ----------- | ----------- |----------- |----------- |----------- |----------- |----------- |----------- |----------- |----------- |
> | F1 improvements    | -0.28       | 0.42 |  0.29 | 0.79| 0.62 | 0.83 | 1.23 | 1.45 | 1.48 | 1.62 |
>
> The pearson correlation is 0.961.
>
> Based on our analysis above, it is evident that as the similarity between relations increases, our method yields more substantial improvements in F1 accuracy.
> We will add a detailed tabular analysis of the correlations between cosine similarity and F1 score improvements in the final version.
> ## Q4: Is there a post-processing step to filter invalid rationales?
> We apologize for not including this due to space limitation. When generating the rationale, we use the following method to filter. Regarding the generated rationale, we also employ the LLM in reverse to classify the relation based on the rationale. In cases where we get incorrect results, we conduct a re-generation of the rationale through the LLM.
>
> We hope our answer can resolve your doubts. Please kindly reconsider the review based on our answer.
>
> [1] Learning robust representations for continual relation extraction via adversarial class augmentation. EMNLP 2022.

---

### Official Review · Reviewer_HVe5 · 2023-08-04

**Typos Grammar Style And Presentation Improvements:** 1.	Line 043-044
**Soundness:** 4

**Excitement:**

3: Ambivalent: It has merits (e.g., it reports state-of-the-art results, the idea is nice), but there are key weaknesses (e.g., it describes incremental work), and it can significantly benefit from another round of revision. However, I won't object to accepting it if my co-reviewers champion it.

**Paper Topic And Main Contributions:**

For continual relation extraction, this paper advocates for the strategy “RationaleCL” which addresses the gap in the models’ capability to infer new relations which are analogous to the already seen relations. Specifically, RationaleCL is a question-answering based supervised technique having three subtasks: question to answer, question to rationale-answer and question-rationale to answer. Moreover, rationale is replaced with “contrastive rationale” to get more context while training the model. Experiments on 2 datatsets: FewRel and TACRED, demonstrate the effectiveness of the strategy.

**Questions For The Authors:**

1. Line 078-083: The authors are requested to clarify this paragraph which requires more context around memory.
2. Line 113-114: The authors are requested to provide more context around memory so that the reader does not have to refer to previous work. A suggestion: Please provide brief overview of Han et al., 2020 in the appendix.
3. Line168-180: Please specify how many (top-k) analogous relations per relation are considered. For example, in Figure1, 2 analogous relations (“follows” and “characters”) are considered for relation “followed by”.
4. Please include some future directions and any follow-up efforts in the conclusion section.

**Reasons To Accept:**

1.	The paper proposes the strategy “RationaleCL” which outperforms the strong baselines.
2.	The paper presents ablation analysis and a case study (Appendix C).

**Reasons To Reject:**

Incomplete ablation analysis in Table2. According to Line177-178 the purpose of cont. replay is to produce contrastive rationale (cri) and replace rationale ri in memory. Now, ri is involved in 2 tasks: Taskr and Taskd . When ri is replaced with cri in both the tasks, there should be 2 “new” tasks for cri. Let’s call them Taskr-cri and Taskd-cri

In Table 2, ablation study is conducted only for following strategies: w/o con. replay and w/o Taskd . However, a complete ablation analysis should include Taskr which is based on ri and one of the missing Tasks based on cri . Moreover, currently, simply mentioning “con. replay” in Table2 creates confusion about which task based on cri is being referred to.

**Reproducibility:**

4: Could mostly reproduce the results, but there may be some variation because of sample variance or minor variations in their interpretation of the protocol or method.

**Reviewer Confidence:**

4: Quite sure. I tried to check the important points carefully. It's unlikely, though conceivable, that I missed something that should affect my ratings.

---

> ### Author Rebuttal · Authors · 2023-08-26
>
> Thanks for your insightful comments! Please see our reponses below.
> ## Q1. The ablation study is not complete.
> Thank you for your advice. We have done the additional ablation study and report the results in the following table.
>
>  |  Models      | TACRED | FewRel |
> | ----------- | ----------- | ----------- |
> | RationaleCL | 80.8 | 85.1 |
> | w/o Taskd-cr | 80.3 | 84.9 |
> | w/o Taskr-cr | 80.6 | 84.9 |
> | w/o Taskd-cr & Taskr-cr | 80.2 | 84.8 |
>
> Conclusively, the ablation study underscores the importance of both Taskd-cr and Taskr-cr components in enhancing the performance of the RationaleCL model. Their presence contributes to the model's robustness to analogous relations, indicating their roles in tackling the catastrophic forgetting challenges of continual learning effectively.
>
> ## Q2. Clarification of memory-related statements in the article.
> We will provide more details around memory in Line 078-083 and Line 113-114, which will be included in our final version.
>
> To alleviate catastrophic forgetting in CL, previous rehearsal-based methods adopt an episodic memory module to store a few representative instances for each previous label. The memory data will be continually learned with emerging new tasks. When a new task is learned, we replay the memory contents associated with prior tasks to prevent forgetting. To select these representative instances for each relation, we use the K-means algorithm to conduct clustering based on the features of all samples extracted by T5-encoder, and select the samples in the centre of each cluster into memory. This approach can effectively preserve the essential label characteristics to address catastrophic forgetting.
>
> ## Q3. How many top-k analogous relations per relation are considered?
> We use cosine similary to judge analogous relations and set the threshold to be 0.97. We consider two relations
> are similar if their similarity exceeds 0.97.
>
> For TACRED dataset, the total relation number is 40. The maximum number of analogous relations per relation is 5. The minimum is 2 and the average is 2.9.
>
> For FewRel dataset, the total relation number is 80. The maximum number of analogous relations per relation is 4. The minimum is 1 and the average is 2.2.
>
> ## Q4. Please include some future directions and any follow-up efforts in the conclusion section.
> Firstly, our method uses manually designed prompts to generate rationales, which may have a large impact on the quality of rationales. Investigating the impact of different prompts on the quality of rationales is an important direction for future research.
>
> Secondly, we currently only use generated rationale to avoid confusion among similar relations in continual learning. In the future, we can use the rationale generated by the LLMfor more complex tasks such as code completion, taks planning, etc.
>
>
> We hope our answers can resolve your doubts. Please reconsider the review based on our answers.

---

### Official Review · Reviewer_hR9i · 2023-08-06

**Soundness:** 3

**Excitement:**

3: Ambivalent: It has merits (e.g., it reports state-of-the-art results, the idea is nice), but there are key weaknesses (e.g., it describes incremental work), and it can significantly benefit from another round of revision. However, I won't object to accepting it if my co-reviewers champion it.

**Paper Topic And Main Contributions:**

The paper investigates continual relation extraction, i.e. learning to extract relations in a continual learning setting. It proposes a method for distilling knowledge from LLMs to perform this task. The method uses reasoning produced by the LLMs for training smaller models. The paper finds that the proposed method outperforms previous methods on two CRE benchmarks. In addition, the paper reports some ablation analyses on the model.

**Reasons To Accept:**

- The approach proposed in the paper is interesting, and it is worth investigating whether LLM-generated reasoning can be used to supervise smaller models in this setting.

- The method seems technically sound

- The experiments are appropriate.

**Reasons To Reject:**

The paper does not provide sufficient details about the experimental results. In particular, it is hard to assess whether the proposed method actually improves over previous methods.

- The paper states that the method outperforms alternatives according to hypothesis tests, with "p<0.5". It is not clear what was tested, or how the data was analyzed. In addition, 0.5 is a very high type I error rate, which would typically be interpreted as consistent with noise.

- The paper does not provide standard deviations or confidence intervals for any of its measurements.

- The ablation results look marginal, and are difficult to interpret without more statistical analysis.

- It is confusing how GPT-3.5 performed poorly on the tasks (much worse than existing baselines apparently), but was able to provide a useful supervision signal for outperforming the baselines. This seems like it should be unlikely. The paper should provide more explanation for what could be going on.

**Reproducibility:**

3: Could reproduce the results with some difficulty. The settings of parameters are underspecified or subjectively determined; the training/evaluation data are not widely available.

**Reviewer Confidence:**

3: Pretty sure, but there's a chance I missed something. Although I have a good feel for this area in general, I did not carefully check the paper's details, e.g., the math, experimental design, or novelty.

---

> ### Author Rebuttal · Authors · 2023-08-26
>
> Thank you for your feedback and helpful suggestions on our work. We hope that the following response could clarify your main concerns.
> ## Q1: How the significance test is performed？
> Firstly, we apologize for the typo "p<0.5" in the paper. In fact, we calculated p<0.05, but we incorrectly wrote p<0.5. Due to space limitation, we did not provide details of the significance test. We will refine the paper by adding the following details.
>
> We use a significance test to determine whether the accuracy of our model is significantly higher than other models after learning the 10th task (T-test, p<0.05). The null hypothesis is that there would be no discrepancies. Specifically, FewRel-ACA: p=0.03; FewRel-CEAR:p=0.001; TACRED-ACA: p=0.002; TACRED-CEAR:p=0.009. Therefore, we can consider that our model significantly outperforms other baseline models. We will detail the significant test results in the final version.
> ## Q2: The paper does not provide standard deviations or confidence intervals for any of its measurements.
> Thanks for your question. We would like to provide some details as follows.
> For the results in Table 1, on FewRel dataset, the standard deviation of our method RationaleCL is:
>
> |  Models      |T1 | T2 | T3 | T4 | T5 | T6 | T7 | T8 | T9 | T10 |
> | ----------- | ----------- | ----------- |----------- |----------- |----------- |----------- |----------- |----------- |----------- |----------- |
> | RationaleCL   |  0.46     | 1.27 | 1.67 | 1.53| 1.06 | 1.03 | 0.69 | 0.74 | 0.51 | 0.23 |
>
> For the results in Table 1, on TACRED dataset, the standard deviation of our method RationaleCL is:
>
> |  Models      |T1 | T2 | T3 | T4 | T5 | T6 | T7 | T8 | T9 | T10 |
> | ----------- | ----------- | ----------- |----------- |----------- |----------- |----------- |----------- |----------- |----------- |----------- |
> | RationaleCL   | 0.97    | 3.15 | 1.99 | 2.97| 2.60 | 1.71 | 1.76 | 1.35 | 0.43 | 0.72 |
>
> We will provide the standard deviations results in the final version.
> ## Q3: The ablation results look marginal, and are difficult to interpret without more statistical analysis.
> Thanks for your question. We would like to provide some details for the significance test of the ablation results.
> The null hypothesis is that there would be no discrepancies between the result of Rationale and other ablation settings and we report  p-value of Paired-Samples T Test.
>
>  |  Models      | TACRED | FewRel |
> | ----------- | ----------- | ----------- |
> | w/o con. replay | 0.022 | 0.016 |
> |  w/o Taskd | 0.025 | 0.024 |
> |w/o both | 0.010 | 0.006 |
>
> We believe these results (p<0.05) can demonstrate the effectiveness of our proposed rationale enhanced framework
> ## Q4: How GPT-3.5 performed poorly on the tasks but was able to provide a useful supervision signal?
> This is a good question. We will explain the reasons below. In fact, our RationaleCL is a multi-task learning model composed of three tasks, including question to answer as the main task, question to rationale-answer
> and question-rationale to answer as auxiliary tasks (see Section 3.2). In this architecture, the LLM-generated rationale  plays an important role in the  two auxiliary tasks to further promote the performance of the main task and the ablation experiments can reflect this (see Table 2.).
>
> The poor performance of GPT-3.5 on relation extraction tasks has been verified in several previous work [1][2]. According to their studies, ChatGPT is limited by the output format requirements in accomplishing fine-grained relation extraction tasks, and it is difficult to directly generate the target relation label within the defined range. However, ChatGPT's semantic understanding is sufficient, and when we provide chatgpt with the correct relation labels, ChatGPT can understand the meaning of the relation according to the context and give reasonable rationales. According to the human-check results[1], **domain experts highly approve of the reasons given by ChatGPT**.
> Here are two cases in our experiment datasets.
>
> 1. For a sentence in the TACRED dataset
>
> Political newcomer **andre carson** is the democratic nominee in a march 11 special election to succeed his late grandmother, **julia carson**, representing indiana's 7th district.
>
> The  relation between "julia carson" and "andre carson" is "person other family". However, LLM predicts the relation to be "person grandparents", which is out of range.  Although the model predicts out-of-scope relations, it actually understands the meaning of the sentences and can give rationales.
>
> 2. For a sentence in the FewRel dataset
>
> During the Crimean War, **Sir George Cathcart** held a dormant commission to take command of the **British Army** in Crimea if Lord Raglan was killed or disabled.
>
> The relation between "Sir George Cathcart" and "British Army" is "military branch", but LLM predicts "position held". However, when given the label in the context, LLM provides reasonable rationale "Sir George Cathcart was a high-ranking military officer and his commission was to take command of the British Army in Crimea, indicating that he was a part of the British Army's hierarchy and military branch."
>
> Thanks again for the useful comments. We will continue to improve our paper according to your suggestions.
>
> [1] Evaluating ChatGPT's Information Extraction Capabilities: An Assessment of Performance, Explainability, Calibration, and Faithfulness. Arxiv
>
> [2] Is Information Extraction Solved by ChatGPT? An Analysis of Performance, Evaluation Criteria, Robustness and Errors. Arxiv

---

### Meta-Review · Area_Chair_qcWy · 2023-09-19

**Recommendation:** 3

**Metareview:**

This paper tackles the problem of continual relation extraction, particularly the problem of inferring new relations which have not appeared so far, but are analogous to the already seen relations. There approach is based on treating this as a question-answering problem and using LLMs for this task. The problem is interesting. All of the reviewers raised some concerns about the evaluation and raised the issue that the approach does not show great improvements over the considered baselines. However, I think the reviewers addressed this somewhat in the author response. I would urge the authors to focus on this aspect in the next version.

---

### Decision · Program_Chairs · 2023-10-07

**Decision:**

Accept-Main

**Comment:**

This paper tackles the problem of continual relation extraction, particularly the problem of inferring new relations which have not appeared so far, but are analogous to the already seen relations. There approach is based on treating this as a question-answering problem and using LLMs for this task. The problem is interesting. All of the reviewers raised some concerns about the evaluation and raised the issue that the approach does not show great improvements over the considered baselines. However, I think the reviewers addressed this somewhat in the author response. I would urge the authors to focus on this aspect in the next version.